# Morphological Evaluation of the Splenic Artery, Its Anatomical Variations and Irrigation Territory

**DOI:** 10.3390/life13010195

**Published:** 2023-01-09

**Authors:** Serghei Covantsev, Fariza Alieva, Karina Mulaeva, Natalia Mazuruc, Olga Belic

**Affiliations:** 1Department of Research and Clinical Development, Botkin Hospital, 125284 Moscow, Russia; 2Department of Surgery, Russian Medical Academy of Continuous Professional Education, 125993 Moscow, Russia; 3Department of Human Anatomy, State University of Medicine and Pharmacy “N. Testemitanu”, 2004 Chisinau, Moldova

**Keywords:** splenic artery, anatomy, polar arteries, accessory spleen, pancreas

## Abstract

Background: Precise knowledge of the topographic features of the splenic artery and its branches in the hilum region is of practical interest due to the various interventions on the vessels of the spleen. Materials and methods: The anatomy of the spleen was studied by means of macroscopic dissection on 330 organ complexes, which were carefully documented and analyzed statistically. Results: The analysis of the splenic artery trajectory led to identification of four types: straight (43.03%), sinusoidal (27.58%), serpentine (20.91%) and alternating (8.48%). To assess the relation between the trajectory of the splenic artery and its branches we performed a chi square test. Sinuous or serpentine trajectory was associated with the presence of long splenic artery branches (dorsal pancreatic artery or the great pancreatic artery), X2 (2, *N* = 330) = 12.85, *p* = 0.001. The artery was located suprapancreatic in 70.30% of cases, anteropancreatic in 4.55%, the vessel had an intrapancreatic course in 14.85% and in 10.00% of cases the artery was located retropancreatic. The presence of inferior polar arteries was associated with a longer pancreas (Spearman’s correlation; r = 0.37; *p* = 0.037). In a multiple regression analysis, inferior polar arteries predicted the length of the pancreas although only a small number of cases could be explained by this model (R2 = 0.127, Adjusted R2 = 0.098; Betta = 0.357; t(330) = 2.091; *p* = 0.045). There were 30 (9.09%) cases of accessory spleens. Conclusions: The arterial supply of the spleen is highly variable in its trajectory, terminal branches, and relation to other organs. The splenic artery tends to be sinuous or serpentine in zones when a large artery branches off (e.g., the dorsal pancreatic or greater pancreatic artery). Multiple short branches tend to stabilize the trajectory of the splenic artery. Inferior polar arteries and accessory spleens contribute to the length of the pancreas, most likely due to increased vascular supply to the tail of the gland.

## 1. Introduction

The spleen is positioned between the fundus of the stomach and the diaphragm in the left hypochondriac region, and this organ shows considerable variations in shape and size [1]. It is supplied by the splenic artery, which most commonly originates from the celiac trunk. The spleen has several important functions in the organism, which are in the main hematological and immune. Due to its structure and location (particularly at the level of ribs IX-XI), it is frequently injured during trauma [2]. The incidence of splenectomy is approximately 6.4–7.1 per 100,000 people per year, while trauma and hematological disorders are the most common cause [3].

The overall mortality in case of splenic trauma is 7–18% and injury to the vascular structures is one of the major risk factors [4,5]. Lacerations, cuts, and dissections of the splanchnic arteries lead to high mortality rates in cases when they are not detected in time [6]. The presence of developmental variations of the vessels can lead to an increase in the duration of surgery as well as increased blood loss.

The study of individual vascular variations is an important method that leads to a decrease in the risk of intra- and post-operative complications, as well as iatrogenic injuries. Although there have been multiple publications about the spleen, there is little data on the variants of the structure of the splenic artery, as studied by means of dissection on a sufficient number of specimens; this method allows the researcher to obtain the most objective information about the structure, size, mass of the organ, and the tissue surrounding it.

Precise knowledge of the topographic features of the splenic artery and its branches in the hilum region is of practical interest due to various interventions on the vessels of the spleen in case of abdominal trauma, portal hypertension, treatment of hypersplenism and splenic tumors [7].

The relevance of this topic is dictated by the introduction of organ-sparing principles of surgical treatment of diseases and injuries of this organ, including arterial embolization, partial or total laparoscopic splenectomy, and endovascular and laparoscopic treatment of splenic artery aneurysms [8,9].

The aim of the current study is to evaluate the morphology of the splenic artery, its variations of development, and its possible origin.

## 2. Materials and Methods

The anatomy of the spleen was studied by means of macroscopic dissection. The study was performed on 330 organ complexes donated to the department of human anatomy by patients who died of causes not related to splenic diseases. The organs were fixed in a 10% formalin solution for 48 h and then carefully dissected. All measurements of the spleen, pancreas, and splenic artery were carefully documented. The spleen was measured in its largest segments. The pancreas was measured from its beginning at the duodenum to the end of its tail. The trajectory of the splenic artery was assessed based on its course and the presence of loops. Polar arteries were defined as vessels which arise from the splenic artery and enter one of the poles of the spleen. The number of branches was calculated based on the number of terminal branches (first order branches) that entered the hilum of the spleen. Branching patterns were classified based on the number and angle of terminal branches. A schematic representation of the main classifications mentioned in the article is presented in Figure 1. The study was conducted according to the ethical laws of the institution and was approved by the ethical commission of State University of Medicine and Pharmacy N.Testemitanu (19.08.2018 nr. 80). The data obtained were analyzed by means of descriptive statistics; data distribution was assessed by the Shapiro-Wilk test, correlation was assessed by Spearman’s analysis and, when possible, by linear regression analysis and the relationship between qualitative data by the chi square test. A *p* value of less than 0.05 was considered statistically significant.

## 3. Results

Variations of the trajectory, course and branching of the splenic artery were studied on 330 specimens. The Shapiro-Wilk test demonstrated a normal distribution of the data (W(330) = 0.98, *p* = 0.08), with skewness of−0.13 ± 0.25 and kurtosis of 0.34 ± 0.55. The splenic length was 9.43 ± 3.62 cm, splenic width 6.71 ± 2.72 mm, and the pancreas length was 17.23 ± 2.41.

The analysis of the splenic artery trajectory led to identification of four types: straight, sinusoidal, serpentine and alternating (a combination of the abovementioned). These types of trajectory are presented in Figure 2, Figure 3 and Figure 4.

We divided the type of arterial supply of the pancreas into three types. In type one, the pancreas was vascularized only by the short branches of the splenic artery; this was encountered in 112 cases (33.94%). In type two, the pancreas was vascularized by the long (dorsal pancreatic artery and/or greater pancreatic artery) and short branches of the splenic artery; this was encountered in 140 cases (42.42%). In type three, the pancreas was vascularized only by the long branches of the splenic artery; this was encountered in 78 cases (23.64%).

To assess the relation between the trajectory of the splenic artery and its branches, we performed a chi square test. Sinuous or serpentine trajectory was associated with the presence of long splenic artery branches (dorsal pancreatic artery and/or the great pancreatic artery), X2 (2, *N* = 330) = 12.85, *p* = 0.001.

The artery was located suprapancreatic in 70.30% (232 cases), anteropancreatic in 4.85% (16 cases), the vessel had an intrapancreatic course (in the parenchyma of the gland) in 14.85% (49 cases) and in 10.00% of cases the artery was located retropancreatic (33 cases) (Figure 5 and Figure 6).

However, the exact location of the splenic artery depends on its course. The course of the splenic artery can be easily assessed in case of a straight trajectory. However, in the case of a sinuous, serpentine or alternating trajectory, it is difficult to single out a specific type. In these types, the trajectory was assessed judging by the prevailing percentage of the trajectory (Table 1).

In the region of the hilum, the splenic artery branched into two arteries of the first order in 273 cases (82.73%), into three arteries in 32 cases (11.11%), and in 11 cases (3.33%), the artery entered the hilum of the spleen without branching. In 10 cases, the vessel branched into four arteries of the first order (3.03%). In rare cases, there were five or six branches of the first order (0.61%, two cases each) (Table 2, Figure 7).

In 67.88% of cases (224 specimens) the artery branched at an acute angle, and in 106 specimens (32.12%), at an obtuse angle. We encountered three predominant types of branching of the artery (Figure 8). The most common type was the bifurcation type (or Y-type)—found in 43.94% (145 specimens) (Figure 8A). In 36.06%, the artery branched as distributed type (119 cases) (Figure 8B). In 66 specimens (20%) the artery had a magistral type of branching (Figure 8C). The splenic artery had intra-arterial anastomoses in 13 cases (3.93%) (Figure 8D).

Polar arteries were defined as vessels that arise from the splenic artery or its branches and vascularize one pole of the spleen. The origin of polar arteries was from the trunk of the splenic artery, or from one of the arteries of the first or second order. In 171 cases (51.82%) there was one superior polar artery (Figure 9A). One inferior polar artery was encountered in 41 cases (12.42%) (Figure 10A). In 10 cases (3.30%), there were two superior polar arteries, with both superior polar vessels branching at different levels of the splenic artery (Figure 9B). The presence of two inferior polar arteries was encountered in 7.27% (24 cases) (Figure 10B). Three superior polar arteries were encountered in 17 cases (5.15%). Both inferior and superior polar arteries were found in 16 cases (4.85%).

The presence of inferior polar arteries was associated with a longer pancreas (Spearman’s correlation; r = 0.37; *p* = 0.037). In a multiple regression analysis, inferior polar arteries predicted the length of the pancreas, although only a small number of cases could be explained by this model (R2 = 0.127, Adjusted R2 = 0.098; Betta = 0.357; t(330) = 2.091; *p* = 0.045).

There were 30 (9.09%) cases of accessory spleens. In the majority of cases the accessory spleen was supplied by terminal branches of the splenic artery. In twelve cases (3.64%) the accessory spleen was supplied by an inferior terminal branch of the splenic artery, in eleven cases (3.33%) by the superior terminal branch of the splenic artery, in four cases (1.21%) from the left gastroepiploic artery, and in three cases (0.91%) by a separate artery directly from the splenic artery. The presence of accessory spleens was associated with a longer pancreas (Spearman’s correlation; r = 0.39; *p* = 0.02).

## 4. Discussion

The number of surgical interventions performed on the spleen and its vessels is constantly increasing. Embolization or ligation of the vessels of the spleen is widely used in case of portal hypertension, oncological diseases, and in various hematological diseases (pancytopenia, thrombocytopenia, hemolytic anemia, Banton’s syndrome, etc.) [7]. All of these procedures are based on a deep understanding of both the anatomy of the organ and the region as a whole. Knowing the variants of the splenic artery is particularly important for visceral surgeons, due to its frequent involvement in gastrointestinal bleeding, organ transplantation, trans-arterial chemoembolization of neoplasms, infusion therapy and iatrogenic injuries [10]. 

The splenic artery is the largest artery of the celiac trunk [9]. Branching from the trunk at an acute angle, it turns to the left, participating in the blood supply of the pancreas, spleen, stomach and their ligaments [5]. 

The trajectory of the splenic artery is variable. There are three main types of trajectories: straight, sinuous and spiral [10]. One or more loops are observed in up to 83–86% of spleens [11,12]. It is assumed that tortuosity increases with age. This is confirmed by the fact that such a trajectory is not typical for the vessels in fetuses, newborns and children, but prevails in the elderly [13,14]. However, one of the studies of the tortuosity of the splenic artery did not reveal a relationship with age or body weight, but instead indicated a relationship between the female sex and a high degree of tortuosity. The authors suggested that this is due to the peculiarity of the influence of female sex hormones on the walls of blood vessels [15]. However, Brinkman and coworkers did not find any relationship between splenic artery loops and age or sex but determined that the length of the artery in contact with the pancreas decreased with age [12]. Michels suggested that the artery is straight in infants and children, minimally tortuous in middle age, and markedly tortuous in the elderly. He suggested that the tortuosity of splenic artery enables the motion of the spleen and allows expansion of the stomach without obstructing blood flow within the splenic artery when passing through the stomach bed. Other possibilities included movement of the spleen with respiration, the ability of the artery to stretch, the damping system developed to provide protection the splenic structure, growth of an artery tethered by its pancreatic branches, and developmental justification. [16,17]. Based on the results of our study the splenic artery is sinuous or serpentine predominantly in the regions where a large arterial vessel branches off to the pancreas. The artery is more commonly straight in cases of multiple small arteries that supply the pancreas (short branches), which tend to stabilize the artery.

Overall, tortuous splenic artery seems to be a benign condition. However, one study showed that extremely tortuous splenic artery was associated in several cases with a clinical disease pattern that resembled chronic pancreatitis [13]. The distal part of the pancreas receives several large arterial branches from the splenic artery and some of the pathogenesis of acute pancreatitis are related to vascular stasis and dysfunction [18,19].

According to our study, the splenic artery had a straight trajectory in 142 cases (43.03%), a tortuous (sinusoidal) trajectory in 91 cases (27.58%), in 69 cases (20.91%) the trajectory was serpentine.

The analysis of topographic relationship of the splenic artery and the pancreas allow it to be classified it into four types [20]. The first type is the suprapancreatic course of the splenic artery, when the artery is located above the pancreas, and this can be seen in 63–99% of cases. The second type is when the artery is located retropancreatic, and it can be encountered in 8–36% of cases. The rarest types are intrapancreatic and anteropancreatic course, which, according to the literature occur in 4–14% and 18.5%, respectively (Table 3) [21,22,23,24,25]. There is also a simplified surgical classification proposed by Wada Y et al. (2020), according to which the splenic artery is divided into two main anatomical types: type S, which curves and passes over the pancreas, and type D, which passes directly and behind the pancreas [26]. A comparison of the available data and the current study is presented in Table 3.

In our study, the vessel was located suprapancreatic in 70.30% (232 cases), and anteropancreatic in 4.55% (16 cases); the vessel had an intrapancreatic passage (in the parenchyma of the gland) in 14.85% (49 cases), and in 10.00%, the artery was located retropancreatic (33 cases). This indicates the prevalence of the suprapancreatic and intrapancreatic course of the splenic artery. The spatial relationship between the splenic artery and the pancreas should be considered in terms of the type of trajectory.

The splenic artery branches into its terminal branches in the region of the hilum of the spleen. There are several types of branching of the artery: distributed (the artery branches into many small branches of the first and second order), magistral (the artery gives off 2–3 large branches of the first order) and bifurcation (division in two main branches). In our study, the most common type was the bifurcation type (or Y-type), which was encountered in 43.94% of cases. In rare cases (2.8–16.7%), there is no branching of the artery, and it enters the spleen parenchyma as a single vessel. A comparison of the available data is presented in Table 4 [20,21,24,27,28].

The number of branches of the first order occupies a separate place in the study of the blood supply to the spleen. The splenic artery branches into two arteries of the first order in 63.1–93.34%, three branches in 6.66–13.6%, more than three branches—in 0.6–18.8% of cases and enters into the parenchyma without branching in 2.8–6.9% (Table 5) [20,24,29].

The presence of polar arteries is just as important, both clinically and fundamentally. The superior polar artery is encountered in 6.66–53% of cases, the inferior in 5.05–54%, and the superior and inferior jointly in 6.66–24.4%, depending on the method, sex and study population (Table 5) [20,22,23,24,29,30,31]. The presence of inferior polar artery tends to correlate with the length of the pancreas, although this relation is weak.

In rare cases terminal branches interconnect before entering into the splenic parenchyma. Anatomical studies demonstrate that contrast solution can pass into neighboring compartments through vascular bridges. This is seen in 9.4% of cases when injecting into secondary branches, 22% when injecting into tertiary branches and 15.66% into fourth order branches [33]. In our study we encountered such anastomoses in 3.93% of cases. Such anatomical variations are especially important in cases of embolization when the material can pass through the anastomoses into a different compartment of the spleen or in the tail of the pancreas.

Michel, while describing the splenic artery, stated: “the artery is so markedly different that no two patterns are the same” [16,17]. In our opinion, this statement remains relevant today, and is increasingly confirmed by the active study of the blood supply to the spleen.

The main limitation of the study is that we did not include sex as a factor to our analysis. This was not possible, as the organ donations were fully anonymous. We also did not assess the intra-organic vascular particularities as this requires other methods (corrosion casts, arteriograms, etc.).

## 5. Conclusions

The arterial supply of the spleen is highly variable in its trajectory, terminal branches and relation to other organs. The splenic artery tends to be sinuous or serpentine in zones when a large artery branches off (dorsal pancreatic or greater pancreatic artery). Multiple short branches tend to stabilize the trajectory of the splenic artery. Inferior polar arteries and accessory spleens contribute to the length of the pancreas, most likely due to increased vascular supply to the tail of the gland. Intra-arterial anastomoses were present in only 3.93% of cases, which is still important to consider during arterial embolization of the spleen.

## Figures and Tables

**Figure 1 life-13-00195-f001:**
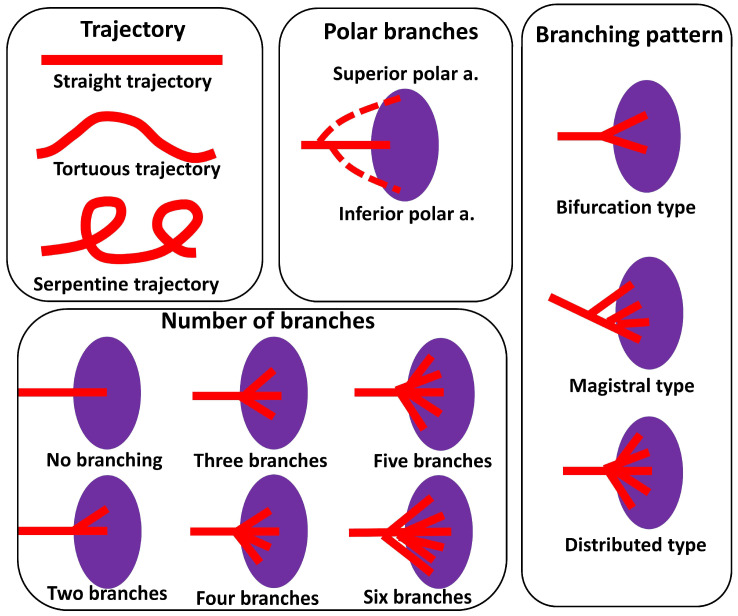
Schematic representation of the classifications mentioned in the article.

**Figure 2 life-13-00195-f002:**
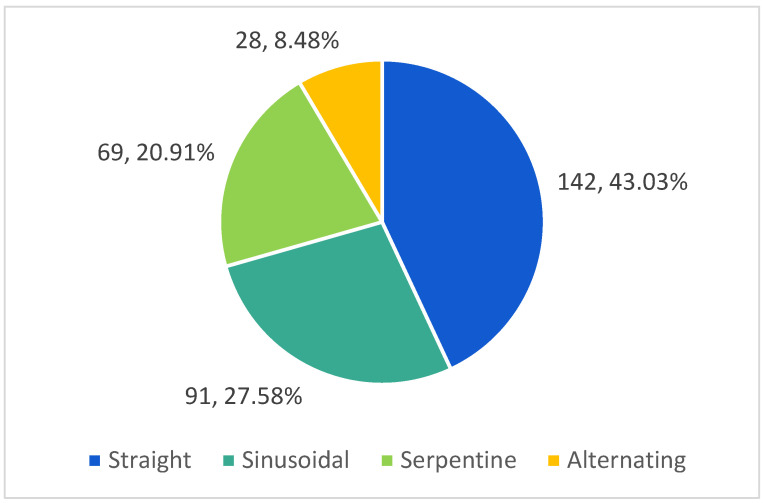
Four types of the trajectory of the splenic artery.

**Figure 3 life-13-00195-f003:**
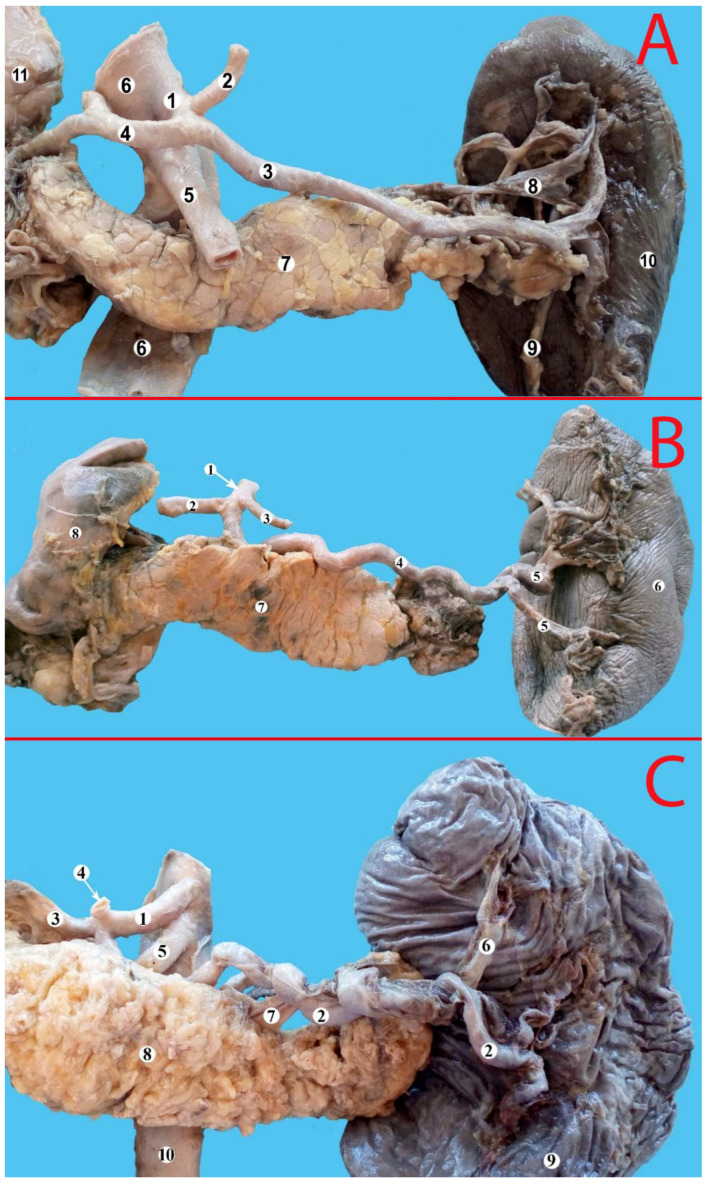
(**A**) Splenic artery with straight trajectory. 1—celiac trunk, 2—left gastric artery, 3—splenic artery, 4—common hepatic artery, 5—superior mesenteric artery, 6—aorta, 7—pancreas, 8—splenic vein, 9—inferior pole artery, 10—spleen, 11—duodenum. (**B**) Splenic artery with a tortuous trajectory. 1—celiac trunk, 2—common hepatic artery, 3—left gastric artery, 4—splenic artery, 5—branches of the first order, 6—spleen, 7—pancreas, 8—duodenum. (**C**) Splenic artery with a serpentine trajectory. 1—celiac trunk, 2—splenic artery, 3—common hepatic artery, 4—left gastric artery, 5—superior mesenteric artery, 6—superior pole artery, 7—splenic vein, 8—pancreas, 9—spleen, 10—aorta.

**Figure 4 life-13-00195-f004:**
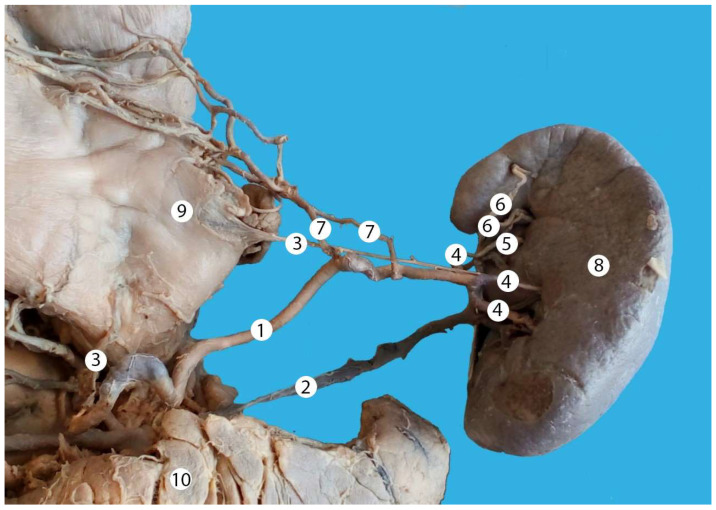
Alternating trajectory of the splenic artery. 1—splenic artery, 2—splenic vein, 3—posterior gastric artery, 4—branches of the splenic artery of the first order, 5—branches of the splenic artery of the second order, 6—superior pole artery, 7—two left gastroepiploic arteries, 8—spleen, 9—stomach (posterior surface), 10—pancreas.

**Figure 5 life-13-00195-f005:**
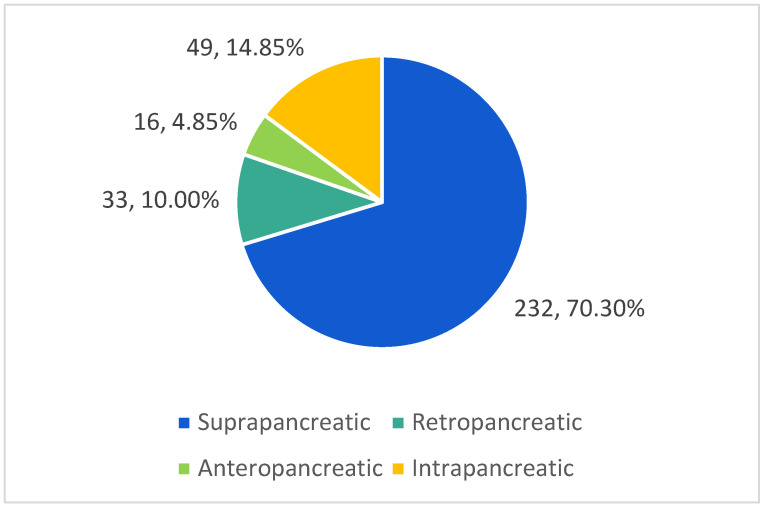
Relation between the splenic artery and the pancreas.

**Figure 6 life-13-00195-f006:**
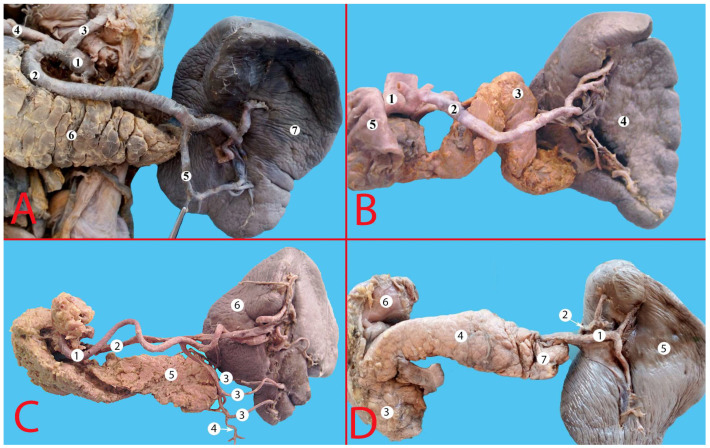
(**A**) Suprapancreatic course of the splenic artery. 1—celiac trunk, 2—splenic artery, 3—left gastric artery, 4—common hepatic artery, 5—inferior pole artery, 6—pancreas, 7—spleen. (**B**) Anteropancreatic course of the splenic artery. 1—aorta, 2—splenic artery, 3—pancreas, 4—spleen, 5—duodenum. (**C**) Intrapancreatic location of the splenic artery (the pancreatic parenchyma is partially removed along the vessel; the vessel is raised up). 1—splenic artery, 2—splenic vein, 3—inferior polar arteries, 4—gastroepiploic artery, 5—pancreas, 6—spleen. (**D**) Retropancreatic course of the vessel. 1—splenic artery, 2—superior polar artery, 3—head of the pancreas, 4—body of the pancreas, 5—spleen, 6—duodenum, 7—tail of the pancreas.

**Figure 7 life-13-00195-f007:**
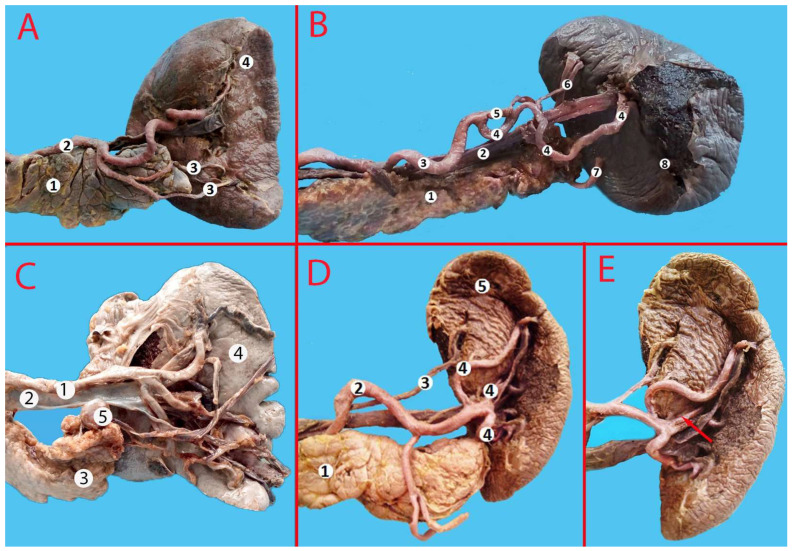
(**A**) The splenic artery enters the parenchyma of the organ without branching. 1—pancreas, 2—splenic artery, 3—inferior polar arteries, 4—spleen. (**B**) Bifurcation of the splenic artery at an obtuse angle. 1—pancreas, 2—splenic vein, 3—splenic artery, 4—branches of the splenic artery of the first order, 5—gastroepiploic artery, 6—superior polar arteries, 7—inferior polar artery; 8—spleen. (**C**) Trifurcation of the splenic artery at an acute angle. 1—splenic artery, 2—splenic vein, 3—tail of the pancreas, 4—spleen, 5—accessory spleen. (**D**,**E**) Division of the splenic artery into 4 branches of the first order. 1—pancreas, 2—splenic artery, 3—polar artery, 4—branches of the splenic artery of the first order, 5—spleen.

**Figure 8 life-13-00195-f008:**
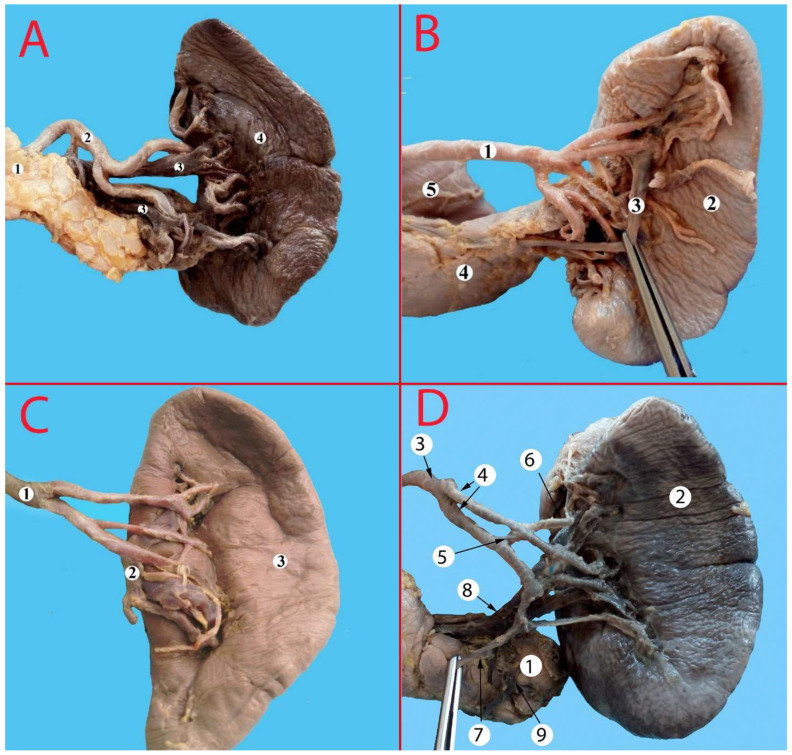
(**A**) Bifurcation type of branching. 1—pancreas, 2—splenic artery, 3—splenic vein, 4—spleen. (**B**) Distributed type of branching. 1—splenic artery, 2—spleen, 3—splenic vein, 4—pancreas, 5—duodenum. (**C**) Magistral type of branching. 1—splenic artery, 2—splenic vein, 3—spleen. (**D**) Intra-arterial anastomosis. 1—pancreas, 2—spleen, 3—splenic artery, 4—branches of the splenic artery, 5—intra-arterial anastomosis, 6—superior polar artery, 7—pancreatic tail artery, 8—splenic vein, 9—vein to the tail of the pancreas.

**Figure 9 life-13-00195-f009:**
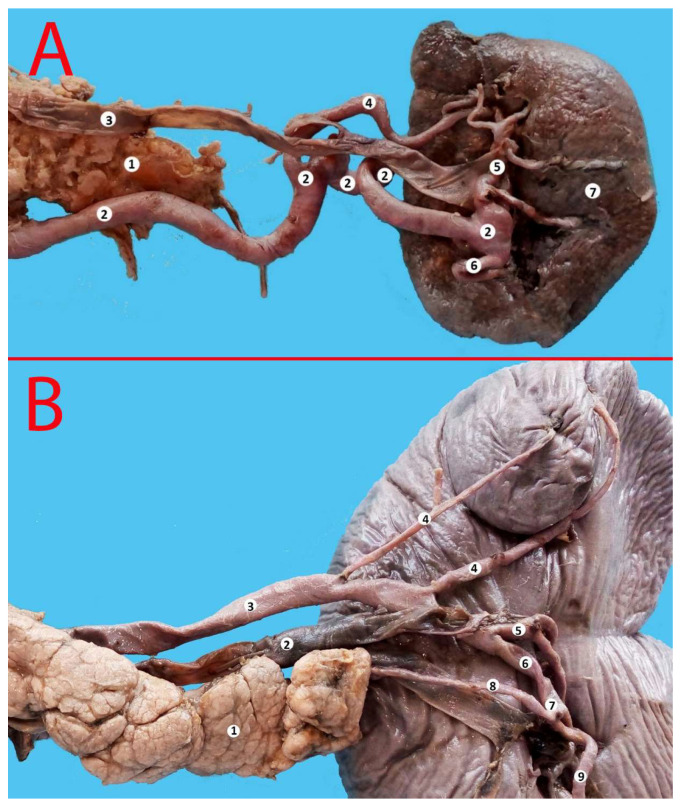
(**A**) Superior polar artery. 1—pancreas, 2—splenic artery, 3—splenic vein, 4—superior polar artery, 5—superior terminal branch of the splenic artery, 6—inferior terminal branch of the splenic artery, 7—spleen. (**B**) Double superior polar artery. 1—pancreas, 2—splenic vein, 3—splenic artery, 4—superior polar artery, 5 superior terminal branch of the splenic artery, 6—inferior terminal branch of the splenic artery, 7—common trunk for the pancreatic tail artery and inferior polar artery, 8—pancreatic tail artery, 9—inferior polar artery.

**Figure 10 life-13-00195-f010:**
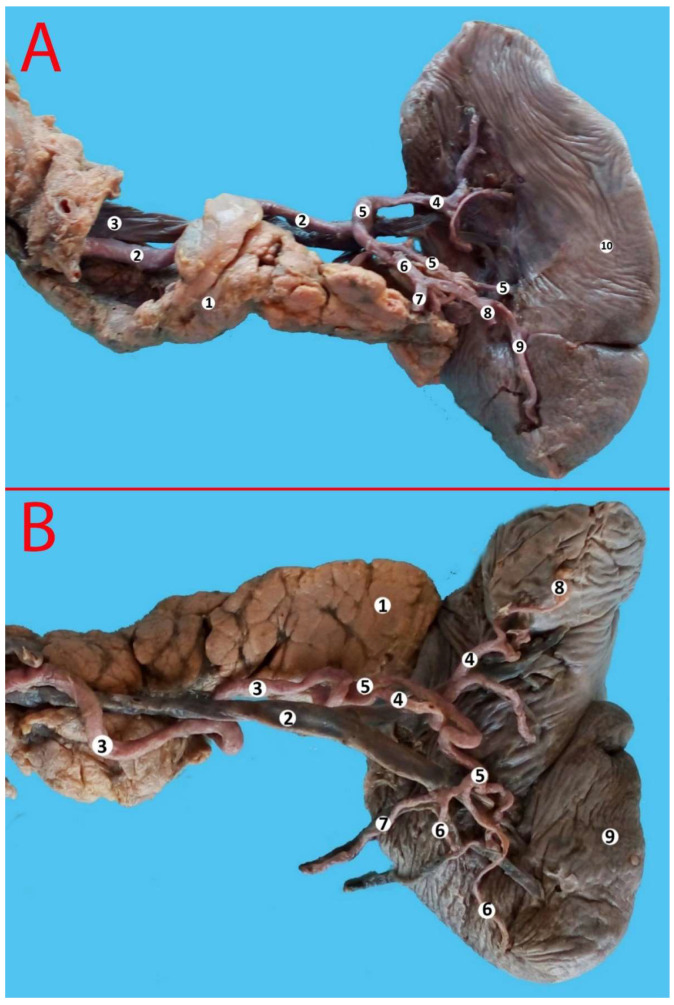
(**A**) Superior polar artery. 1—pancreas, 2—splenic artery, 3—splenic vein, 4—superior terminal branch of the splenic artery, 5—middle terminal branch of the splenic artery, 6—inferior terminal branch of the splenic artery, 7—artery to the tail of the pancreas, 8—gastroepiploic artery, 9—inferior polar artery, 10—spleen. (**B**) Double superior polar artery. 1—pancreas, 2—splenic vein, 3—splenic artery, 4—superior terminal branch of the splenic artery, 5—inferior terminal branch of the splenic artery, 6—inferior polar arteries, 7—gastroepiploic artery, 8—superior polar artery, 9—spleen.

**Table 1 life-13-00195-t001:** Relation between the trajectory and position relative to the pancreas.

	SP	IP	RP	AP	Total
Straight, n (%)	109 (33.03)	15 (4.55)	16 (4.85)	2 (0.61)	142 (43.03)
Sinuous, n (%)	56 (16.97)	15 (4.55)	9 (2.73)	11 (3.33)	91 (27.58)
Serpentine, n (%)	45 (13.64)	17 (5.15)	7 (2.12)	0 (0.00)	69 (20.91)
Alternating, n (%)	22 (6.67)	2 (0.61)	1 (0.30)	3 (0.91)	28 (8.48)
Total, n (%)	232 (70.30)	49 (14.85)	33 (10.00)	16 (4.85)	330 (100)

SP—suprapancreatic, IP—intrapancreatic, RP—retropancreatic, AP—anteropancreatic.

**Table 2 life-13-00195-t002:** Number of terminal branches of the splenic artery.

Number of Branches	*n* (%)
No branching	11 (3.33)
Two branches	273 (82.73)
Three branches	32 (9.70)
Four branches	10 (3.03)
Five branches	2 (0.61)
Six branches	2 (0.61)
Total	330 (100%)

**Table 3 life-13-00195-t003:** Variants of the trajectory of the splenic artery.

Author, Year	Number of Specimens	SP	IP	RP	AP	References
Pandey et al., 2004	320	74.1%	4.6%	8.2%	18.5	[24]
Gangadhara et al., 2014	30	63.3%	-	36.3%	-	[23]
Ashok et al., 2016	76	68%	-	32%	-	[21]
Meet Krishna, 2017	317	99.3%	6.66%	-	-	[22]
Zhu et al., 2018	169	36.7%	14.2%	49.1		[25]
Current study	330	70.3%	14.85%	10%	4.55%	

SP—suprapancreatic, IP—intrapancreatic, RP—retropancreatic, AP—anteropancreatic.

**Table 4 life-13-00195-t004:** Branching pattern of the splenic artery.

Author, Year	Number of Specimens	No Branching	Magistral Type	Distributed Type	References
Pandey et al., 2004	320	2.8%	97.2		[24]
Xu et al., 2009	48	-	60.9	39.1	[27]
Zheng et al., 2015	317	-	64.7	35.3	[20]
Ashok et al., 2016	76	10.5	55.3	34.2	[21]
Sangeetha, & Sundar, 2020	60	16.7	60	23.3	[28]
Current study	330	3.33	20	36.06	

**Table 5 life-13-00195-t005:** The number of terminal branches and polar arteries.

Author, Year	Number of Specimens, n	Number of Splenic Artery Branches, %				Polar Arteries, %			References
		1	2	3	More than 3	Superior	Inferior	Both	
Liu et al., 1996	850	0.8	86	12.2	1	** 31.3	** 38.8	** 13.3	[32]
Daisy Sahni et al., 2003	200	-	80 (M) 79.5 (F)	20 (M) 19,5(F)	-	53	33	-	[30]
Pandey, 2004	320	2.8%	63.1	-	4–18.8 6–9.7 >6–5.6	-			[24]
Silva et al., 2011	60 + 30 *	-	93.34 90 *	6.66 10 *	-	16% and 20% respectively			[29]
Shashikala Londhe, 2013	50		90	10	-	33	54	24.4	[31]
Gangadhara et al., 2014	30	-	80	16.6	-	26.6	36.6	16.6	[23]
Zheng et al., 2015	317	6.9	78.9	13.6	0.6	16.4	5.05	-	[20]
Meet Krishna, 2017	317	-	100	-	-	6.66	-	6.66	[22]
Current study	330	3.33	82.73	9.70	4.25	60.27	19.69	4.85	

M—males, F—females, * 60—anatomical specimens and 30 contrast enhanced x-rays ** 280 specimens.

## Data Availability

Available at request.

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
