# Peer review of "Morphological Evaluation of the Splenic Artery, Its Anatomical Variations and Irrigation Territory"

_life, 2023, doi:10.3390/life13010195_

Round 1
Reviewer 1 Report
The study "The enigmatic anatomy of the splenic artery" is an interesting attempt at an extended analysis of the relationship between the splenic artery course/variant and branching pattern. This approach is quite innovative and deserves recognition. However, the manuscript in its current version requires some minor corrections.
The sentence "Spleen is an organ which is located in the upper left part of the abdominal cavity and varies in its shape and size" can be rephrased as "Spleen is positioned between the fundus of the stomach and the diaphragm in the left hypochondriac region. However, this organ shows considerable variations in shape and size." Relationships between the Spleen and ribs (IX-XI) should be mentioned because the authors note, "Due to its structure and location it is frequently injured during trauma."
There is much chaos in the labelings of anatomical structures in the figures. The exact anatomical structures are marked with different numbers on each subfigure. Use letter symbols (e.g., LG for left gastric artery) or unified numbers so that a given number always corresponds to the same structure on a given figure.
As the analysis of the splenic artery trajectory allowed authors to distinguish four types of the Spleen: straight, sinusoidal, serpentine, and alternating (a combination of the abovementioned), each type must be characterized in the results. Which criteria determined the qualification of a specific specimen to a given type?
Author Response
Thank you for the time and effort taken to review the manuscript. We hereby provide a step-by-step revision of the comments:
The sentence "Spleen is an organ which is located in the upper left part of the abdominal cavity and varies in its shape and size" can be rephrased as "Spleen is positioned between the fundus of the stomach and the diaphragm in the left hypochondriac region. However, this organ shows considerable variations in shape and size." Relationships between the Spleen and ribs (IX-XI) should be mentioned because the authors note, "Due to its structure and location it is frequently injured during trauma."
-Thank you. We made several changes in the introduction and included this comment.
There is much chaos in the labelings of anatomical structures in the figures. The exact anatomical structures are marked with different numbers on each subfigure. Use letter symbols (e.g., LG for left gastric artery) or unified numbers so that a given number always corresponds to the same structure on a given figure.
-Thank you. The anatomical structures are abundantly labeled. We made several changes, included larger numbers to facilitate reading. However, the figures are different and we cannot label the uniformly in the manuscript.
As the analysis of the splenic artery trajectory allowed authors to distinguish four types of the Spleen: straight, sinusoidal, serpentine, and alternating (a combination of the abovementioned), each type must be characterized in the results. Which criteria determined the qualification of a specific specimen to a given type?
-Thank you, we have extended the material and methods section.

Reviewer 2 Report
The scientific paper "The enigmatic anatomy of the splenic artery” aimed to evaluate the morphology of the splenic artery, its variations of development and its possible origin. I can make the following considerations:
1) Author names and affiliations are incomplete in template. Please adjust to the standards required by the Life MDPI journal;
2) I suggest that authors change the title of the manuscript. The word enigmatic is not a common term in anatomy. I suggest that you change this title. For example: Morphological evaluation of the splenic artery, its anatomical variations and irrigation territory;
3) The abstract is too long. I suggest shortening the results and conclusions section;
4) Needs more details on the materials and methods used to obtain the measurement of the spleen, pancreas and splenic artery;
5) The numbers inserted in figure 3 must be larger, for better visualization;
6) On lines 117 to 120, regarding the location of the artery, it is written: “The artery was located suprapancreatic in 70.30% (232 cases), anteropancreatic in 4.55% (15 cases), the vessel had an intrapancreatic course (in the parenchyma of the gland) in 14.85% (49 cases) and in 10.00% of cases the artery was located retropancreatic (33 cases) (Figure 4, 5)". The sum 232 + 15 + 49 + 33 = 329. Isn't that 330 organ complexes?
7) I recommend reviewing mathematical sums throughout the manuscript.
8) In figure 5C and 5D the letters are very small. Please adjust for better viewing;
9) Insert in Figure 5D a number with legend for the tail of the pancreas;
10) The description of the results is confusing. I suggest using more tables for the description of measured values and reducing the textual description;
11) The discussion tables are not in the Life journal standard (MDPI), as well as the references should be adjusted;
12) Insert possible limitations of the study at the end of the discussion;
13) Include the clinical applicability of the study in the conclusions.
Author Response
Thank you for the time and effort taken to review the manuscript. We hereby provide a step-by-step revision of the comments:
- Author names and affiliations are incomplete in template. Please adjust to the standards required by the Life MDPI journal;
-Thank you for the comment. However, we have to submit the manuscript anonymously, therefore this part is intentionally left blank. We provided all the identifying information in the system.
2) I suggest that authors change the title of the manuscript. The word enigmatic is not a common term in anatomy. I suggest that you change this title. For example: Morphological evaluation of the splenic artery, its anatomical variations and irrigation territory;
-Thank you, we changed the title of the manuscript.
3) The abstract is too long. I suggest shortening the results and conclusions section;
-We made the abstracted shorter
4) Needs more details on the materials and methods used to obtain the measurement of the spleen, pancreas and splenic artery;
-We extended the material and methods section. It now includes more information.
5) The numbers inserted in figure 3 must be larger, for better visualization;
-Thank you. We changed the figure.
6) On lines 117 to 120, regarding the location of the artery, it is written: “The artery was located suprapancreatic in 70.30% (232 cases), anteropancreatic in 4.55% (15 cases), the vessel had an intrapancreatic course (in the parenchyma of the gland) in 14.85% (49 cases) and in 10.00% of cases the artery was located retropancreatic (33 cases) (Figure 4, 5)". The sum 232 + 15 + 49 + 33 = 329. Isn't that 330 organ complexes?
-Indeed, there was 1 case missing. We changed this part and rechecked all of the calculation.
7) I recommend reviewing mathematical sums throughout the manuscript.
-Thank you, we rechecked the calculation.
8) In figure 5C and 5D the letters are very small. Please adjust for better viewing;
-Adjusted
9) Insert in Figure 5D a number with legend for the tail of the pancreas;
-Corrected
10) The description of the results is confusing. I suggest using more tables for the description of measured values and reducing the textual description;
-We do agree that some parts of the manuscript are hard to read. We made the results section shorted and included a table in the most complicated part. Most of the manuscript is now presented in a short and simple fascion.
11) The discussion tables are not in the Life journal standard (MDPI), as well as the references should be adjusted;
-We changed the tables and edited the references
12) Insert possible limitations of the study at the end of the discussion;
-Thank you, the discussion section is supplemented.
13) Include the clinical applicability of the study in the conclusions.
-Thank you, we extended the conclusion section.

Round 2
Reviewer 2 Report
No comments